# Impacts of Air Pollution on Health and Cost of Illness in Jakarta, Indonesia

**DOI:** 10.3390/ijerph20042916

**Published:** 2023-02-07

**Authors:** Ginanjar Syuhada, Adhadian Akbar, Donny Hardiawan, Vivian Pun, Adi Darmawan, Sri Hayyu Alynda Heryati, Adiatma Yudistira Manogar Siregar, Ririn Radiawati Kusuma, Raden Driejana, Vijendra Ingole, Daniel Kass, Sumi Mehta

**Affiliations:** 1Environmental, Climate, and Urban Health Division, Vital Strategies, Singapore 068807, Singapore; 2Center for Economics and Development Studies, Department of Economics, Faculty of Economics and Business, Universitas Padjadjaran, Bandung 40115, Indonesia; 3Environment Agency of DKI Jakarta Province, Jakarta 13640, Indonesia; 4Faculty of Civil and Environment Engineering, Institut Teknologi Bandung, Bandung 40132, Indonesia; 5Environmental, Climate, and Urban Health Division, Vital Strategies, New York, NY 10005, USA

**Keywords:** air pollution, health impacts, economic impacts, cost of illness

## Abstract

(1) Background: This study aimed to quantify the health and economic impacts of air pollution in Jakarta Province, the capital of Indonesia. (2) Methods: We quantified the health and economic burden of fine particulate matter (PM_2.5_) and ground-level Ozone (O_3_), which exceeds the local and global ambient air quality standards. We selected health outcomes which include adverse health outcomes in children, all-cause mortality, and daily hospitalizations. We used comparative risk assessment methods to estimate health burdens attributable to PM_2.5_ and O_3_, linking the local population and selected health outcomes data with relative risks from the literature. The economic burdens were calculated using cost-of-illness and the value of the statistical life-year approach. (3) Results: Our results suggest over 7000 adverse health outcomes in children, over 10,000 deaths, and over 5000 hospitalizations that can be attributed to air pollution each year in Jakarta. The annual total cost of the health impact of air pollution reached approximately USD 2943.42 million. (4) Conclusions: By using local data to quantify and assess the health and economic impacts of air pollution in Jakarta, our study provides timely evidence needed to prioritize clean air actions to be taken to promote the public’s health.

## 1. Introduction

Global evidence on the adverse health impacts of air pollution is consistent and clear; however, there is limited local evidence of the burden of air pollution and its associated monetary costs in Jakarta, Indonesia. Air pollution is a major threat to more than 10.5 million people’s health in Jakarta. Based on the data from the Environmental Agency of Jakarta Province, the annual ambient PM_2.5_ concentrations in Jakarta are the highest among all urban centers in Indonesia. The World Health Organization (WHO) has stated that air pollution is one of the major environmental risks to health, leading to both morbidities and mortalities, including cancers, cardiovascular diseases, and respiratory diseases [1]. In 2019, the Global Burden of Disease (GBD) Study estimated that air pollution caused 5054 deaths (or 54 per 100,000 people) and 168,000 years lost to ill-health, disability, or premature death in Jakarta [2].

Air pollution has been strongly linked to non-communicable diseases (NCDs), including cardiovascular and chronic respiratory diseases and lung cancers [3], which impose substantial burdens on the healthcare sector and the economy of the country [4]. In Jakarta, NCDs accounted for 79% (36,000 deaths) of total deaths in 2019 [5]. By causing NCDs and premature death, air pollution contributes to productive labor loss and increased health care expenditures, reduces the country’s gross domestic product (GDP), makes cities less productive and competitive, and lowers the quality of life of its residents. Recently, the World Bank reported that air pollution in Indonesia is attributable to an annual cost of over USD 220 billion (6.6% of Indonesia’s GDP (PPP)) in 2019 [6].

Recognized by the WHO as “an overlooked health emergency for children around the world”, air pollution can be severe, especially for children living in low- or middle-income countries, where air quality levels far exceed the WHO’s health-based guidelines [7]. Children have special risks from air pollution since their organs (e.g., heart and lungs) and systems (e.g., respiratory and cardiovascular) are still developing. In addition, they breathe in more air per kilogram of body mass because they have higher respiration rates compared to adults [8]. There is also growing evidence linking air pollution to child stunting and adverse birth outcomes such as low birth weight and preterm birth [9,10]. Developmental setbacks in children from these early-life outcomes have prolonged impacts throughout their lives.

Air pollution-related economic impacts within the health arena include healthcare costs (e.g., treatment costs) as well as short and long-term costs associated with illness (e.g., income loss). Studies have shown that the cost of treating illness due to pollution in Asia can reach USD 5.4–USD 9.1 billion [11]. Non-healthcare costs from air pollution’s impact on children can be extremely high given the lifelong impacts of exposure [12]. Similarly, premature deaths from air pollution also result in substantial economic impacts [13].

Health impact assessment is an approach to assess potential health impacts for public health improvement and policy-making purposes [14]. By applying this approach, our study addresses an important health assessment in children attributable to air pollution, as the information regarding the health impacts in children is lower compared to other age groups or vulnerable populations. The latest GBD 2019 Study only includes low birth weight and small for gestational age as the adverse health outcomes in children attributable to PM_2.5_ exposure [3]. The current study fills the information gap by assessing more health impacts of air pollution in children, i.e., infant deaths, childhood stunting, low birth weight, and premature birth, and by using officially published local data. Furthermore, to understand the magnitude of health and economic burdens attributable to air pollution in Jakarta, we aim to estimate the health impacts due to PM_2.5_ and ground-level Ozone (O_3_), focusing on adverse health outcomes in children, mortality, and daily hospitalizations. The economic impacts associated with these health burdens are also quantified. The results of our current study could be leveraged to provide timely scientific evidence to support the formulation of clean air policies in Jakarta.

## 2. Materials and Methods

### 2.1. Study Area

The study area covered five districts in the Special Capital District (DKI) of Jakarta Province: Central, North, South, West, and East Jakarta, where the air quality monitoring stations are located. Figure 1 shows the administrative division of the DKI Jakarta province.

### 2.2. Exposure Data (PM_2.5_ and O_3_ Concentration)

We collected air pollution data for 2018 and 2019 from the Environmental Agency of Jakarta Province. We focused on health-damaging pollutants that routinely exceed daily and annual levels set in the Indonesian National Ambient Air Quality Standards (NAAQS), namely PM_2.5_ and O_3_. Since the larger particulate matter PM_10_ includes PM_2.5_, we select only PM_2.5_ to avoid double counting. The selection of the pollutants of interest is consistent with other approaches to estimating the health burdens from ambient air pollution [3,15,16].

Daily pollutant data from five air quality monitoring stations in five districts: Central Jakarta, North Jakarta, South Jakarta, East Jakarta, and West Jakarta, were compiled. The data were transformed into annual mean and daily mean concentrations (for PM_2.5_), daily maximum 8 h average, and annual mean of daily maximum 8 h average concentrations (for O_3_). Data on PM_2.5_ in West and East Jakarta were unavailable until late 2020; prior to that, only PM_10_ was monitored. To account for the missing PM_2.5_ data, the ratio of PM_2.5_ to PM_10_ was calculated from the available data in each area and used to estimate PM_2.5_ concentrations using PM_10_ concentration data for 2019. These ratios were assumed to remain constant throughout the year.

We applied a conversion factor of 0.51 to convert the concentration level of O_3_ in µg/m^3^ to part per billion (ppb) as recommended by the US EPA (1 ppb of O_3_ = 1.97 µg/m^3^ at 298K and 1013 mbar) [17]. This conversion was necessary due to the different units used in the concentration-response functions and the concentration levels obtained from the monitoring stations.

To ensure consistency with the methodology used in the GBD 2019 Study, the annual average PM_2.5_ exposure was calculated by also adding the average concentration of household air pollution (negligible given that all households in Jakarta have already converted to clean household energy) to the annual average ambient PM_2.5_ concentration in Jakarta [3].

### 2.3. Health Data

The study focused on the exposures and outcomes in 2019, except for the number of hospitalizations from 2018, the most recent year for which data were available from the Indonesia Health Care and Social Security Agency (BPJS). Baseline birth and population data at the district and provincial levels were collected from the publicly available data from the National Central Bureau of Statistics (BPS). Data on the prevalence of stunting, prevalence of infant mortality, prevalence of newborns with low birth weight, and prevalence of preterm births were obtained from the Health Agency of Jakarta Province. All health data other than daily hospitalization data in 2019 were provided by the Health Agency as part of its routine monitoring of the city. The daily hospitalization data has already been de-identified by the BPJS to ensure data privacy protection and confidentiality of the patients maintained.

Consistent with the comprehensive review of the available epidemiology used within the comparative risk assessment framework in the GBD Study 2019, the following health outcomes were selected for inclusion in the health impact assessment. Adverse health outcomes in children: infant deaths and adverse birth outcomes (i.e., preterm births and newborns with low birth weight); Mortality: total mortality and six cause-specific mortality (i.e., due to ischemic heart disease (IHD), due to chronic obstructive pulmonary disease (COPD), due to stroke, due to lung cancers, due to type 2 diabetes mellitus, and due to lower respiratory infection (LRI)). We calculated the total mortality caused by the six specific diseases by multiplying the total population of Jakarta in 2019 obtained from the BPS with the mortality rates estimated by the GBD 2019 Study. Stunting was included based on its priority as a key children’s outcome of concern in the country, with risk based on a recently published meta-analysis [10]. Daily hospitalizations due to two cause-specific hospital admissions (i.e., due to cardiovascular diseases and due to respiratory illnesses) were also included, with risks recommended by a comprehensive meta-analysis study conducted in Europe [18].

### 2.4. Estimation of the Health Burdens of Air Pollution

#### 2.4.1. Long-term Impacts of Air Pollution

In this study, the long-term impacts of air pollution were defined as the health burdens attributable to annual exposure to air pollution. Health outcomes addressed include adverse outcomes in children and total mortality, including six cause-specific mortalities. Table 1 presents the relative risks (RR) used to show that health burdens might evolve due to changes in exposure in the future.

The number of premature deaths (excluding infant mortality) associated with PM_2.5_ pollution for six causes (i.e., ischemic heart disease, stroke, chronic obstructive pulmonary disease, type 2 diabetes mellitus, lung cancer, and lower respiratory infections) were also examined (outcome definition with corresponding ICD-10 codes are summarized in Appendix A).

To estimate cause-specific deaths attributable to PM_2.5_ pollution, the concentration-response (or relative risk) relationship between PM_2.5_ exposure and specific causes of death was based on the new MR-BRT (Meta-Regression—Bayesian, Regularized, Trimmed) RR curves developed in the GBD 2019 analysis [20]. The mean and 95% confidence intervals of the RRs for each disease (and age intervals where necessary) were provided as a look-up table by the GBD collaborators. COPD mortality attributable to O_3_ pollution in Jakarta was estimated using a relative risk of 1.06 (1.03–1.10) per 10 ppb of O_3_ exposure as recommended by the GBD Study 2019 [21].

#### 2.4.2. Short-term Impacts of Air Pollution

Short-term impacts were defined as health burdens attributable to short-term (on the order of days) changes in exposure to air pollution. In this study, total daily hospitalizations due to cardiovascular and respiratory diseases associated with short-term exposures to air pollution were estimated.

Data on daily hospitalizations in 2018 were obtained from the Healthcare and Social Security Agency [22]. ICD-10 codes were used to categorize the cause of hospitalization into a larger group, i.e., I00–I99 for cardiovascular diseases and J00–J99 for respiratory diseases. The relative risk estimates used to obtain the total daily hospitalizations attributable to air pollution are summarized in Table 2.

#### 2.4.3. Calculating Health Burdens Attributable to Air Pollution

A health impact assessment was performed using comparative risk assessment methodology. This approach estimates changes in health burdens attributable to air pollution at a population level from a given exposure level. The RR estimates listed in Table 1 and Table 2 were applied to exposure concentrations to determine the expected number of adverse health events attributable to ambient air pollution. Except for total mortality attributable to PM_2.5_, the cause-specific disease burdens attributable to air pollution were estimated using Equation (1):(1)$$Y_{AP}=D\times \left[1-e^{\left[-\beta \left(c-c_{f}\right)\right]}\right]$$
where:$Y_{AP}$ = Number of health outcomes attributable to air pollutionD = Baseline number of health endpoints over the study periodβ = the coefficient as the slope of the log-linear relationship between ambient air pollution concentrations and health outcomes, or the exponentiation of RR c = baseline PM2.5 concentration (in µg/m3) in a day or in a year (e.g., 2019)$c_{f}$ = counterfactual concentration of pollutants (i.e., 4.2 µg/m3 for PM2.5 and 32.4 ppb for O3), below which no additional health risk is conferred

We used the GBD Study 2019 methodology and MR-BRT RR curves to estimate the cause-specific deaths attributable to PM_2.5_ exposure. The MR-BRT model was developed to statistically describe the nonlinear patterns for the association between PM_2.5_ levels and various diseases. The model was fitted to effect estimates from the latest epidemiologic studies of ambient PM_2.5_ pollution. The counterfactual PM_2.5_ concentration was set to have a uniform distribution from 2.4 µg/m^3^ to 5.9 µg/m^3^ (mean of 4.2 µg/m^3^), while for O_3_, the counterfactual level ranges from 29.1 ppb to 35.7 ppb (mean of 32.4 ppb) [3]. Subtracting the counterfactual levels to the ambient concentration of both PM_2.5_ and O_3_ was required to estimate health burdens attributable to the air pollutants. Figure 2 illustrates the general idea of calculating the health burdens attributable to air pollution and the related economic costs.

#### 2.4.4. Estimation of Economic Impacts of Health Burdens Attributable to Air Pollution

This section describes the detailed approach used to conduct the economic assessment of the study. All the assumptions used to calculate the cost of illness attributable to air pollution are listed in Table 3. We used the 2018 BPJS sample data released in 2020, which contains a sample of BPJS participant data. The sample was randomly selected from the strata of BPJS participants for the 2016–2018 period. The data includes 1. Membership; 2. Visits of BPJS patients who seek treatment at first-level health facilities; 3. Visits of BPJS patients who seek treatment at Advanced Level Referral Health Facilities (FKRTL); and 4. Treatment costs. The distribution of hospital visits in the sample data set was used to generate population-level estimates for the entire Jakarta province. We estimated 7,227,665 outpatient visits and 576,733 inpatient visits in 2018.

#### 2.4.5. Estimating Hospitalization Days

The average inpatient length of stay was calculated from the sample data of BPJS patients who sought treatment at FKRTL. The average length of inpatient days in 2018 for each disease category was calculated for each of the ICD-10 codes used to estimate health impacts.

#### 2.4.6. Estimating Inpatient Health Care Cost

Disease-specific inpatient treatment costs were estimated using the cost information available in BPJS 2018 data. The total treatment costs per year for each disease were divided by the number of hospitalizations for the respective disease to estimate the unit costs of treatment per case per disease. The unit costs were then multiplied by the number of attributable cases calculated for each outcome to obtain the total treatment cost due to air pollution. The nominal value of treatment cost was adjusted using the Indonesian health care cost inflation rate [12].

#### 2.4.7. Estimating Non-Health Care Cost

Patients who undergo inpatient care were assumed to lose productivity during their stay, resulting in non-health care costs. The monthly minimum wage of Jakarta Province was used as a proxy for productivity value. Monthly values were divided by twenty to obtain estimates of daily productivity loss. Daily productivity loss was then multiplied by the average inpatient days for each case per disease to obtain productivity loss per case per disease.

#### 2.4.8. Estimating the Value of Statistical Life Year (VSL)

The estimation of Indonesian VSL is based on the approach developed by Robinson et al. [27], using equation (2):(2)$$VSL_{Indonesia}=VSL_{US}\times {\left(\frac{GNI_{Indonesia}}{GNI_{US_{\;}}}\right)}^{elasticity}$$

Using the elasticity of 1.5, following the same study by Robinson et al. for consistency [27], the 2019 VSL in Indonesia was estimated to be USD 284,184.25. The number of attributable deaths due to pollution was multiplied by the amount of VSL to determine the value of loss due to premature death.

#### 2.4.9. Estimating the Cost of Stunting

The cost of treating stunting was proxied by the cost of interventions to prevent stunting. This followed the study by Hoddinott et al. [12], where the cost of preventing stunting is considered as the value that we put to avoid stunting and, as such, as the value of the disease itself. The cost of preventing stunting in 2013 was adjusted to 2019 values using the Indonesia health care inflation estimate [30] and multiplied by the number of attributable stunting cases due to air pollution to obtain the total cost of stunting prevention.

## 3. Results

### 3.1. Demography and Air Pollution Level in Jakarta

Table 4 describes the demographic characteristics of Jakarta according to the data from Jakarta Provincial Government [31]. East Jakarta has the largest area and the highest population of all cities in Jakarta. However, West Jakarta is the most densely populated area in Jakarta. Even though the poverty line is highest in South Jakarta, the percentage of poor people living in the area is the lowest compared to other cities. In terms of GRDP per capita, Central Jakarta has the highest GRDP per capita, while East Jakarta has the lowest.

As shown in Figure 3, the annual level of PM_2.5_ in Jakarta in 2019 was three times higher than the NAAQS. Similar to PM_2.5_, the annual daily max 8 h average of O_3_ was also almost two times higher than the NAAQS. Meanwhile, the annual level of NO_2_ and SO_2_ was below the requirement set in the NAAQS; therefore, they were omitted from further analyses.

### 3.2. Health Impacts

#### 3.2.1. Long-Term Impacts of Air Pollution

Annual exposure to PM_2.5_ causes 6100 cases of stunting, 330 infant deaths, 700 infants with adverse birth outcomes, as well as nearly 9700 premature mortality (Table 5). The high level of O_3_ caused nearly 310 deaths due to COPD among the population aged 25 years and above (Table 5). District-specific results are described in Appendix A and Appendix A, while the ranges of uncertainty with 95% confidence intervals of these estimates are described in Appendix A.

#### 3.2.2. Short-Term Impacts of Air Pollution

Daily exposure to air pollution was associated with over 5000 cases of hospitalizations in a year. Exposure to PM_2.5_ may lead to nearly 3500 hospitalizations; 87% of PM_2.5_-related admissions were due to cardiovascular diseases. On the other hand, exposure to high-level O_3_ may cause more than 1500 hospitalizations among people aged 65 years and above, of which 83% were due to cardiovascular diseases (Table 6). District-specific results are presented in Appendix A, while the ranges of uncertainty with 95% confidence intervals of these estimates are described in Appendix A.

#### 3.2.3. The Economic Cost of Health Impacts from Air Pollution

The total cost per year of the health impacts from air pollution reached approximately USD 2943.42 million, equivalent to 2.2% of Jakarta Province’s GDRP. The summary of the cost (in billion USD) at the provincial level is provided in Table 7, while the estimates at the city level are presented in Appendix A.

## 4. Discussion

This is the first comprehensive study to evaluate health and economic burdens attributable to air pollution in Jakarta Province, combining the results of GBD 2019 with local health and economic data. We found that each year, air pollution causes more than 10,000 deaths, 5000 hospitalizations for cardio-respiratory diseases, and more than 7000 adverse outcomes in children resulting in an economic cost of approximately USD 2943.42 million (2.2% of Jakarta Province GRDP).

The total mortality in the current study was two-fold higher than GBD 2019 estimates, mainly due to different air pollution levels and the number of people exposed to these exposures. A limited number of studies have been conducted on the health and economic burdens attributable to air pollution in Jakarta. However, the studies mainly looked at acute symptoms and incidence of diseases attributable to air pollution, such as asthma attacks, respiratory symptoms, the incidence of COPD, incidence of pneumonia, incidence of bronchitis, and incidence of other respiratory illnesses [34,35,36,37]. A study by Resosudarmo & Napitupulu predicted 7900 deaths due to air pollution in 2015 [35]. The estimate was lower compared to our estimation, which could be due to differences in the methods applied to estimate the total mortality attributable to air pollution. In 2012, the Breathe Easy Jakarta study estimated that the mortality caused by air pollution was 3700 for 20 µg/m^3^ of PM_2.5_ [37]. In other words, when the annual PM_2.5_ level reached 52 µg/m^3^, the total mortality attributable to PM_2.5_ was 9620, which is slightly lower than our estimate. Factors that may contribute to this difference are the methods applied to determine the exposure level, pollutants of interest, and concentration-response functions used to estimate the mortality attributable to air pollution.

Our study has shown that the economic loss due to PM_2.5_ and O_3_-related deaths and illness is around 2.2% of Jakarta Province’s GDRP. This is lower than the national estimates calculated by The World Bank, which had estimated the economic loss of PM_2.5_-related health damages in 2019 to be USD 220 billion (6.6% of Indonesia’s GDP (PPP)) [6], and OECD, which found the economic loss to be USD 96.4 billion (or 3.5% of Indonesia’s economy) in 2015 [38]. However, the more prominent issues are the permanent effects of some of the health impacts.

The economic impact of deaths comprises the largest share of the total cost, and this value has already reflected the value of productivity loss in relation to deaths [39]. Although this cost is already high, we have not included the potential multiplier effect (e.g., the larger effect that consumption brings to the economy [40]) that might occur due to loss of consumption due to premature deaths. As such, the impact is likely higher than our estimate. In addition, our future productivity loss estimation stemming from stunting reveals a high economic loss. Stunted children might feel such an impact throughout their productive years. Although some interventions, such as nutrition, zinc, and vitamin supplementation, may reduce the magnitude of the impact, the effect is largely irreversible [41,42]. Therefore, a more aggressive approach to pollution control is urgently needed.

Moreover, this study highlights that East Jakarta city has the highest economic loss, amounting to USD 790.94 million in 2019, in line with the severity of the health impact. The mortality cost of around 11 billion Rupiah is about one-fourth of the total mortality cost in the entire Jakarta Province. East Jakarta city also has the highest cost of stunting and hospitalization, warranting more attention.

Significant findings from our current studies are consistent with other studies assessing health and economic burdens from air pollution. In 2021, the Public Health Agency of Canada reported that air pollution was attributed to 15,300 premature deaths per year, which led to an economic cost of CAD 114 billion [43]. Meanwhile, in Thailand, the mortality burdens attributable to air pollution were estimated to be 50,000 deaths per year, with an economic cost of USD 60.9 billion (almost 15% of Thailand’s GDP) in 2016 [44]. Similarly, recent studies also showed the striking impacts of air pollution on health and related economic costs [16,45,46], suggesting that air pollution is a major threat to environmental health around the globe.

Recently, the WHO published a new AQG that generally has set more stringent limit values for each pollutant to ensure maximum protection of human health from exposure to air pollution. The annual limit value for PM_2.5_ and O_3_ set in the recent WHO AQG (5 µg/m^3^ for PM_2.5_; 60 µg/m^3^ for O_3_ during peak season) [33] is similar to the theoretical minimum-risk exposure level (counterfactual level) determined by the GBD 2019 Study (2.4–5.9 µg/m^3^ for PM_2.5_; 57.3–70.3 µg/m^3^ for O_3_) [3]. A study conducted in Europe showed that more than 50,000 deaths per year attributable to PM_2.5_ could be avoided if European cities complied with the recent AQG [47].

It should be noted that an important assumption used in this study is that people living in the same area had the same level of exposure to PM_2.5_ and O_3_. Therefore, the findings presented in the current study can only be interpreted as burdens at a population level, not an individual level. The O_3_-related findings should also not be extrapolated beyond the age group from which the risk estimates have been generated. Also, the air pollution assessment used in this study is limited to areas where the monitoring stations have been installed with limited detection coverage. The results may not be representable to the whole area of Jakarta Province as Kepulauan Seribu does not have a monitoring station installed and therefore has not been included in the analyses. Since Jakarta has a limited number of air quality monitoring stations, a statistical model accounting for trends, seasonality, and other meteorological factors should be considered in future studies to fully characterize the spatial and temporal distribution of air pollution in the entire province [48,49].

Furthermore, the number of hospitalizations was limited to active members of national insurance (BPJS). According to the statistics bureau of Jakarta Province, 84% of the Jakarta population was registered as active BPJS members in 2018 [50]. For this reason, the study findings, especially the attributable burden estimates for cause-specific mortality and the number of hospitalizations, may have been underestimated.

Other possible pollutants, such as NO_2_ and SO_2_, were not included in this study, and exposure to PM_2.5_ and O_3_ might not be sufficient to fully characterize the toxicity of the atmospheric mix or to fully account for the risk of mortality associated with exposure to ambient pollution. However, the possible burden from those pollutants is expected to be comparatively small, given their low ambient concentrations. In addition, our estimates of attributable burdens of air pollution depend on a number of assumptions, as well as the rates and the relative risks obtained from epidemiological studies, all of which, if changed, would alter the estimates. Despite these limitations, our study provides useful up-to-date estimates of health and economic loss attributable to ambient air pollution in Jakarta through 2019 using the most recent estimates from the GBD 2019 Study and comprehensive local health data.

## 5. Conclusions

We estimated that air pollution potentially caused more than 10,000 deaths, more than 5000 hospitalizations for cardio-respiratory diseases, and more than 7000 adverse health outcomes in children each year in Jakarta. The total economic burden attributable to air pollution was estimated to be USD 2943.42 million (2.2% of its GRDP) for 2019. By using local data to quantify and assess the health and economic impacts of air pollution in Jakarta, both on mortality and adverse health outcomes in children, our study provides timely evidence needed to guide city policymakers as they prioritize clean air actions to be taken to promote the public’s health.

## Figures and Tables

**Figure 1 ijerph-20-02916-f001:**
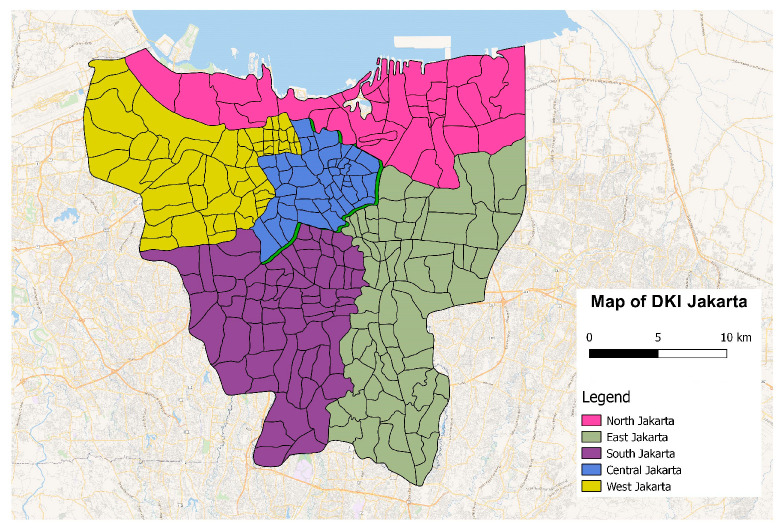
Map of Administration Division of the DKI Jakarta Province.

**Figure 2 ijerph-20-02916-f002:**
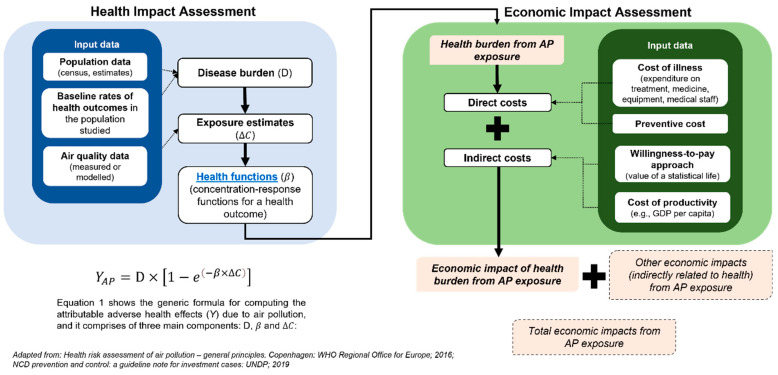
Assessment framework used to quantify the health burdens and economic burdens attributable to air pollution. The ΔC is the subtraction result of the recent PM_2.5_ level (c) and the counterfactual PM_2.5_ $\left(c_{f}\right)$ selected to calculate the health impacts.

**Figure 3 ijerph-20-02916-f003:**
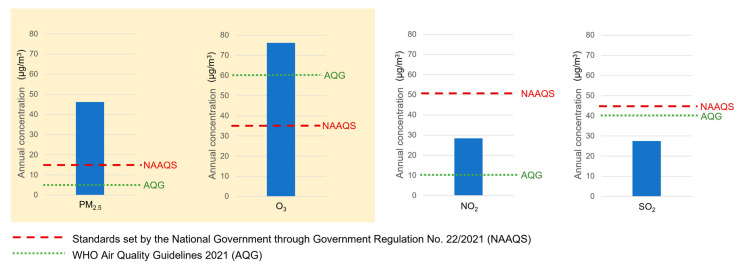
Air pollutants were prioritized based on exceedances of the National Ambient Air Quality Standards [32] and the World Health Organization’s Air Quality Guidelines 2021 [33]. Air pollutants of interest are highlighted in cream.

**Table 1 ijerph-20-02916-t001:** Relative risks (RR) and 95% confidence interval (CI) for children and birth outcomes for a 10 µg/m^3^ change in PM_2.5_ exposure.

Health Endpoints	Age	RR (95% CI)
Infant Mortality, all-cause	1–12 months	1.09 (1.04, 1.14) [19]
Stunting	<5 years old	1.19 (1.10, 1.29) ^1^ [10]
Low Birth Weight at term	At birth	1.18 (1.06, 1.33) [9]
Preterm Birth	At birth	1.007 (1.005, 1.08) [9]

^1^ RR estimate for household air pollution, with the assumption that underlies the impact of household air pollution, is the same as the impact of ambient air pollution.

**Table 2 ijerph-20-02916-t002:** Relative risks (RR) and 95% confidence intervals (CIs) for total hospitalizations for a 10 µg/m^3^ change in PM_2.5_ and O_3_ exposure.

Total Hospitalizations	Age	RR (95% CI)
*For PM_2.5_*		
Due to Cardiovascular diseases Due to Respiratory diseases	All ages All ages	1.0091 (1.0017, 1.0166) [18] 1.0019 (0.9982, 1.0402) [18]
*For O_3_*		
Due to Cardiovascular diseases	65+	1.0089 (1.0050, 1.0127) [18]
Due to Respiratory diseases	65+	1.0044 (1.0007, 1.0083) [18]

**Table 3 ijerph-20-02916-t003:** Assumptions used for economic impact estimates.

No	Assumption	Amount
1	Exchange Rate USD 2019 (rounded) [23]	IDR 14,000
2	Jakarta Monthly Minimum Wage 2019 [24]	USD 281.50
3	GNI Indonesia Per Capita 2019 (current) [25]	USD 4051.78
4	GNI United States Per Capita 2019 (current) [25]	USD 66,061
5	VSL (value of statistical life) United States 2019 [26]	USD 10,900,000
6	VSL Indonesia 2019 (adjusted) [27]	USD 284,184.25
7	Income Elasticity [27]	1.5
8	Purchasing Power Parity (USD–IDN) [25]	IDR 4.75
9	Productive Age [28]	15–64
10	Productive Years	49 Years
11	GDP Jakarta 2019 (In Billion USD) [29]	USD 131.64
12	Healthcare Inflation (2014–2019) [12]	5.71% (2014); 5.32% (2015); 3.92% (2016); 2.9% (2017); 3.14% (2018); 3.46% (2019)
13 14 15 16 17	Low Birth Weight * Cost Preterm Birth * Cost Stunting *^ Cost Hospital Admission for CVD * Cost Hospital Admission for RESP * Cost	USD 1513.03 USD 1136.17 USD 128.26 USD 1313.83 USD 638.98

* Calculation based on BPJS Data Sample; ^ Calculation based on Hoddinott et al. [12].

**Table 4 ijerph-20-02916-t004:** Demographic characteristics of Jakarta Province and the five cities.

Characteristics	Jakarta Province *	Central Jakarta	North Jakarta	West Jakarta	South Jakarta	East Jakarta
Total Area (km^2^)	664	52.4	140	124.4	154.3	182.7
Number of Population	10,557,810	928,109	1,812,915	2,589,933	2,264,699	2,937,859
Population Density (per km^2^)	15,900	17,719	12,950	20,813	14,675	16,080
Poverty Line (IDR/capita/month)	667,260	625,177	549,506	517,646	729,256	539,510
Number of Poor People (% within the municipality) ^a^	365,550 (3.5%)	34,130(3.7%)	91,090 (5.0%)	84,020 (3.2%)	61,760 (2.7%)	91,610 (3.1%)
GRDP per capita (in USD)	19,056.71	53,886.57	20,592.21	12,976.36	20,267.86	11,880.43
Land-use Characteristic	Mixed	Governmental Center	Industrial & Port	Industrial (small-scale)	Residential	Industrial

* Note that Jakarta Province also includes Thousand Islands as a municipality, but it is not considered in this study; ^a^ Poor people are residents with an average expenditure per capita per month below the poverty line [31].

**Table 5 ijerph-20-02916-t005:** Long-term impacts of air pollution on mortality and adverse outcomes in children.

Health Outcomes	Total burden	Air Pollution Attributable Indicators	
		Number of Cases	Rate
**PM_2.5_ (annual mean: 52 µg/m^3^)**			
*Adverse outcomes in children (2019)*			
Infant Deaths	986	327	2 per 1000 births
Stunting	11,211	6153	7 per 1000 children under 5
Low Birth Weight	1269	680	5 per 1000 births
Preterm Births	1919	62	4 per 10,000 births
*Mortality (2019) **	23,430 ^	9692	88 per 100,000 population
**O_3_ (annual daily max 8 h average: 274 µg/m^3^)**			
Mortality due to COPD (2019) *	3635	310	5 per 100,000 aged 25+

* Mortality count is calculated by multiplying the Jakarta population with the mortality rate obtained from GBD 2019 Study results. ^ Total mortality includes deaths from ischemic heart disease, stroke, COPD, type 2 diabetes, lower respiratory infections, and lung cancer.

**Table 6 ijerph-20-02916-t006:** Short-term impacts of air pollution on daily hospitalizations.

Hospitalizations	Disease Cause	
	Cardiovascular	Respiratory
*PM_2.5_*		
Total Hospitalizations	150,272	108,560
PM_2.5_ Attributable Indicators		
Number of Cases	3043	455
Rate (per 100,000 population)	28	4
*O_3_ **		
Total Hospitalizations	37,039	12,147
O_3_ Attributable Indicators		
Number of Cases	1357	182
Rate (per 100,000 population *)	282	38

* O_3_-related outcomes are only for the population aged 65 years and over.

**Table 7 ijerph-20-02916-t007:** The annual economic cost of health impact attributable to air pollution in Jakarta Province (in a million USD).

	Adverse Health Outcomes in Children			Mortality	Hospitalizations
	Infant Deaths	Stunting	Adverse Birth Outcomes *		
Health Burden	327	6153	742	10,002	5037
Economic Cost	92.93	0.79	1.10	2842.41	6.19

* Combining newborns with low birth weight and preterm birth.

## Data Availability

The data that support the findings of this study are available from datasets provided by the Environmental Agency of DKI Jakarta (data on air quality), Indonesia Health Care and Social Security Agency (data on daily hospitalizations), and Health Agency of DKI Jakarta, but restrictions apply to the availability of these data, which were used under license for the current study, and so are not publicly available. Data are, however, available from the authors upon reasonable request and with permission of the Environmental Agency of DKI Jakarta and Indonesia Health Care and Social Security Agency.

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
