# Peer review of "Impacts of Air Pollution on Health and Cost of Illness in Jakarta, Indonesia"

_ijerph, 2023, doi:10.3390/ijerph20042916_

Round 1

Reviewer 1 Report

I found this topic to be extremely informative, and I was very impressed with the way in which the authors presented their findings. Following are a few minor observations and comments that may help to further enhance your research work. The authors should address the following points in detail.

The abstract should describe the paper's accomplishments.

_ The introduction of the paper should include the contribution more prominently.

_ A “Literature Review” section/subsection should be added to the paper. The literature review section should include a tabular comparison of existing models to identify their shortcomings and strengths. The literature section is lacking in the enclosure of formulation and pollution aspects and requires more recent studies to be reviewed. Therefore, I suggest you cite 2022 and 2021 papers including the following works to get an idea of paper structure formulation and analysis:

 https://doi.org/10.3390/atmos12101338

https://doi.org/10.3389/fenvs.2022.945628

 -Define the preliminary in the methodology section to explain all the mathematical notations and symbols used in the paper.

-It is necessary to improve the quality of the figures.

-The authors should consider the comparison of results with existing studies.

- Formatting should be seriously revised by the authors.

Reviewer 2 Report

I usually am very critical, particularly, when it comes to the professionalism of presentation, scientific soundness and language, but honestly I have been pleased with this article from the first moment I laid my eyes upon it. This is one of the top first-draft manuscripts I have reviewed in the last 5 years in terms of quality of presentation, language and scientific robustness.  For that, I thank the authors for their careful and high quality work.

I therefore recommend its publication in its current form, but I encourage the authors to do one more final check for minor typos and language issues e.g.

 Line 23 – 24 : The sentence is broken and there is unnecessary comma before ‘and’:

Please use either of my suggestions:

We found that over 7,000 adverse health outcomes in children, over 10,000 deaths and over 5,000 hospitalizations can be annually attributed to air pollution in Jakarta.

Or

We estimated over 7,000 adverse health outcomes in children, over 10,000 23 deaths and over 5,000 hospitalizations that can be attributed to air pollution annually in Jakarta.

Please also remove all the commas before ‘and’ and ‘or’ e.g. Lines 38-42

Line 93: focused not focus

Thank you,

Author Response

We thank the referee for his/her constructive suggestions. Detailed point-by-point responses to all comments and suggestions are provided below.

Comment B1: I usually am very critical, particularly, when it comes to the professionalism of presentation, scientific soundness and language, but honestly I have been pleased with this article from the first moment I laid my eyes upon it. This is one of the top first-draft manuscripts I have reviewed in the last 5 years in terms of quality of presentation, language and scientific robustness.  For that, I thank the authors for their careful and high quality work.

Response B1: We thank the referee for the positive feedback and encouragement.

Comment B2: I therefore recommend its publication in its current form, but I encourage the authors to do one more final check for minor typos and language issues e.g. line 23 – 24 : The sentence is broken and there is unnecessary comma before ‘and’:

Response B2: We thank the referee for the observation and now we have removed the comma before “and”. We have also done a careful check for minor typos/language.

Comment B3: Please use either of my suggestions:

We found that over 7,000 adverse health outcomes in children, over 10,000 deaths and over 5,000 hospitalizations can be annually attributed to air pollution in Jakarta.

Response B3: We thank the reviewer for the suggestion and have revised it for clarity as follows:

“Our results suggest over 7,000 adverse health outcomes in children, over 10,000 deaths, and over 5,000 hospitalizations that can be attributed to air pollution each year in Jakarta.”

Comment B4: Please also remove all the commas before ‘and’ and ‘or’ e.g. Lines 38-42

Response B4: We thank the referee for the observation. We have removed all the commas before “and” and “or” accordingly.

Comment B5: Line 93: focused not focus

Response B5: We thank the referee for the observation, and we have revised it accordingly.

We thank the referee for such positive comments and a very insightful review.

Reviewer 3 Report

In this manuscript, a very important study has been conducted to quantify the health and economic impacts of air pollution in Jakarta Province, the capital of Indonesia. Such impacts were derived from data representing particulate matter and ground level ozone concentrations. The other variables included adverse health outcomes in children, all-cause mortality, and daily hospitalizations. The research utilised  comparative risk assessment methods to estimate health burdens attributable to PM2.5 and O3 linking local population and mortality data with PM2.5–mortality relationships from the literature. As a result, very interesting results have been generated and their discussion has been carried out as well. The manuscript also reports on the strengths and weaknesses inherent in the conceptual design of the study. For example, other possible pollutants such as NO2 and SO2 were not included in this study. The manuscript continues further to mention that exposure to PM2.5 and O3 might not be sufficient to fully characterize the toxicity of the atmospheric mix or to fully account for the risk of mortality associated with exposure to ambient pollution. My view is that to circumvent this study limitation, there is a need for more historical data on the atmospheric pollutants under consideration. Currently, we are told that air pollution data were collected for the year 2018 and year 2019 only, which is too brief a period. My view is that there is a need for at least 5 year data sets or longer to estimate the infant mortalities. In other words, there is a need to compare at least two different scenarios and then show their different or similar mortality levels. Otherwise how do we determine the cause-effect relationships in a statistically convincing manner. Therefore, the authors can demonstrate mortalities 5 years ago (period 1)  and mortalities during the 2018-2019 period (period 2). Alternatively, the methodology can compare infant mortalities in areas with less atmospheric pollution levels against infant  mortalities in highly polluted areas.

Round 2

Reviewer 1 Report

My concerns have been addressed by the authors. Other than that, I do not have any questions

Reviewer 3 Report

I have read the new inputs and further explanations given in the revised manuscript. Having gone through them, I am satisfied that my concerns or comments  have been addressed and thus the paper has been adequately reinforced and is now more balanced.